



# Role of black carbon in the formation of primary organic aerosols: Insights from molecular dynamics simulations

Xiaoqi Zhou[1], Yulu Zhou[1], Sylvain Picaud[2], Michel Devel[3], Jesús Carrete[4], Georg K. H.  Madsen[4], and Zhao Wang[1,4]

[1]Department of Physics, Guangxi University, 530004 Nanning, China
[2]Institut UTINAM, CNRS UMR 6213, Université Bourgogne Franche-Comté, 25030 Besançon, France
[3]FEMTO-ST Institute, UBFC, CNRS, ENSMM, 15B avenue des Montboucons, 25030 Besançon, France
[4]Institute of Materials Chemistry, TU Wien, 1060 Vienna, Austria

**Correspondence:** Zhao Wang (zhao.wang@tuwien.ac.at)

**Abstract.**
Many studies on the mixing state of suspended particulate matters (PM) have pointed to the role of carbon particles as
nucleation seeds in the formation of atmospheric aerosols. However, the underlying physicochemical mechanisms remain
unclear, particularly concerning the involvement of volatile organic compounds (VOCs) at the primary stage of clustering. Here
we gain insights into those microscopic formation mechanisms through molecular dynamics simulations of the physisorption
of gaseous organic molecules on the surface of a carbon nanoparticle (NP). Six different organic species are selected among the
VOCs dominating the atmospheric pollutants of several megacities, to interact with an onion-shell nanostructure that mimics
the primary soot particle. We consider organic molecules at various densities on the surface of a NP, as well as the same
molecules in the gas phase without any NP.
The pollutant molecules are found to cluster in clearly different ways in the presence of the NP than in the gas phase.
The contrast in the binding energy of molecular clusters confirms the catalytic role of black carbon in the primary formation
of aerosols from VOCs. Morphology analysis reveals different clustering behaviors of aromatic and aliphatic compounds,
leading to differences in the thermal stability of the formed PMs. Our simulations also suggest a layer-by-layer formation
process of aerosol PM, consistent with the onion-like nanostructures of aerosol particles previously observed in transmission
electron microscopy experiments. These results shed light on the microscopic mechanisms of primary aerosol formation, and
are correlated with a variety of experimental measurements on aerosol PMs and VOCs.

## 1 Introduction

Atmospheric aerosol particulate matter (PM) takes part in many environmental processes that impact climate and health, so its
formation has been the object of intensive research efforts. Volatile organic compounds (VOCs) emitted into the atmosphere





from diverse environmental sources have been reported to act as precursors of organic aerosols (Hallquist et al., 2009; Tao
et al., 2017; Volkamer et al., 2009; Carlton et al., 2009; Ziemann and Atkinson, 2012; Gentner et al., 2017; Hettiyadura et al.,
2019; Majdi et al., 2019; Li et al., 2019; Lim et al., 2019; Maclean et al., 2017). Numerical simulations have been carried
out at the molecular level to gain insights into the morphology of aerosols and the energetics of their nucleation from VOCs
(Chakraborty and Zachariah, 2011; Li et al., 2010; Ma et al., 2011; Zhao et al., 2019; Hede et al., 2011). Notably, molecular
dynamics (MD) simulations have been used to study the morphology and clustering of organic and inorganic compound under
atmospherically relevant conditions (Li et al., 2010; Darvas et al., 2011; Ma et al., 2011; Darvas et al., 2013; Radola et al.,
2017; Karadima et al., 2019).
Many studies on the mixing state of aerosol PM have pointed to a role of black carbon (BC) as nucleation seeds for the
formation of aerosol PM (Adachi et al., 2010; Bondy et al., 2018; Chen et al., 2017; Metcalf et al., 2013; Niemi et al., 2006;
Li et al., 2011, 2015; Fu et al., 2012; Zhang et al., 2015; Riemer et al., 2019). BC is usually formed by incomplete combustion
of fuels and biomass, and is often found in soot particles (Dallmann et al., 2014; Nienow and Roberts, 2006). It is abundantly
present in the atmosphere of modern cities, especially in mega-cities in northern China due to coal burning and vehicle emission
(Han et al., 2010; Rose et al., 2011; Cheng et al., 2012; Almanza et al., 2012; Wang et al., 2016; Ueda et al., 2018). Many
experimental efforts have been devoted to understand the role of BC in the formation of aerosol PM and to study its impact
on the visibility reduction and climate change (Koch et al., 2011; Ueda et al., 2016; Forestieri et al., 2018; Mahrt et al., 2018;
Yu et al., 2019; Lefevre et al., 2019). There is therefore clear interest in characterizing the currently unclear physicochemical
processes at the root of aerosol formation, particularly the primary stage of molecular clustering involving the interaction
between BC and VOCs.
Here we use MD simulations to study the physisorption of gaseous organic compounds on BC nanoparticles, which is
correlated to the primary formation process of aerosol PM from VOCs. The binding energy and morphology of the molecular
clusters obtained from molecular simulations are analyzed as a way to gain insight into the role of BC in the primary growth
of organic aerosols. The molecular clusters formed on the NP are found to be energetically mores table than those formed in
the gas phase, which points to a catalytic role of black carbon in the primary formation of aerosols from VOCs. Furthermore,
morphology analysis reveals different values of the binding energy and thus different thermal stability of aromatic and aliphatic
compounds can be related to different ways of clustering.

## 2  Methods

Primary soot particles of BC have often observed to exhibit sub-micron fullerene-like onion-shell structures in experiments
using laser desorption mass spectrometry (LDMS) and transmission electron microscopy (TEM) (Li et al., 2003; Wentzel et al.,
2003; Nienow and Roberts, 2006). To mimic those reported structures, the carbon nanoparticle (NP) in this work is modeled as
a bucky-onion of $3.64\,\mathrm{nm}$ in diameter containing four concentric fullerene layers (Langlet et al., 2007). The system size is kept
small due to computational cost considerations, even though BC particles in urban atmospheres can grow from tens to over a





hundred nanometers after mixing with other compounds through atmospheric aging processes (Rose et al., 2006; Adachi and
Buseck, 2008; Lee et al., 2015; Pei et al., 2018).
VOCs emitted from both biogenic and anthropogenic sources often dominate the atmospheric pollution in megacities (Bar-
letta et al., 2005; Da Silva et al., 2018; Hsieh and Tsai, 2002; An et al., 2011; Ras et al., 2009; Molteni et al., 2018; Cao
et al., 2018; Yang et al., 2018). Here we select six representative carbohydrates from VOCs reported by these previous works
as adsorbates, because of their elemental but diverse structures. These samples include ethylene, propylene, toluene, styrene,
ethylbenzene and para-xylene, all shown in Fig. 1 (a).
The atomistic interactions are described in the framework of the adaptive interatomic reactive empirical bond order (AIREBO)
potential, in which the total energy is built from individual bond contributions involving many-body terms. The long-range van
der Waals (vdW) interactions are included by adding a parametrized *Lennard-Jones* (LJ) function with a cutoff radius of
$1.0\,\mathrm{nm}$. Details concerning the formulation, parameterization and benchmarks of the AIREBO potential are provided else-
where (Stuart et al., 2000). This force field has recently shown good accuracy in describing the deformation and adsorption
behaviors of carbon nanostructures (Wang, 2009; Wang and Philippe, 2009; Petucci et al., 2013; Sun and Bai, 2017).
Like in our earlier works (Wang and Devel, 2011; Wang, 2019a), the adsorption process is simulated by integrating the
equations of motion of all atoms in the system using the parallel MD package Lammps (Plimpton, 1995). Organic molecules
are initially placed at random sites near the surface of NP in a periodic simulation cell of about $10 \times 10 \times 10\,\mathrm{nm}^3$, as shown
in Fig. 1 (b). Initial velocities are sampled from a Maxwell-Boltzmann distribution. A thermostat is then used to let the NP
progressively reach thermal equilibrium at $300\,\mathrm{K}$ in about $0.5\,\mathrm{ns}$, during which most of the molecules interact with the NP
surface; this interval was determined to be enough after testing with the case of toluene as a reference. The case without NP is
also simulated under the same conditions for a purpose of comparison, as illustrated in Fig. 1 (c). In this case, the thermostat is
applied directly to the molecules instead.
A repeated heating-annealing process is used to compute the statistical energy of atomization of the system. Energy mini-
mization is performed via an annealing process after reaching thermal equilibrium, in order to take a "frozen" picture of the
system, from which the energy is calculated. After optimization, the temperature is raised again and molecules are free to
move at $300\,\mathrm{K}$. This heating-annealing process is repeated for 13 times (determined after a convergence test) in each case to
let the system hop among metastable states and compute an average. The full set of Lammps inputs required to replicate these
simulations is provided with the online version.
A key coefficient influencing the clustering of molecules, the per-molecule binding energy $\varepsilon$ is calculated as the difference
in the energy of atomization between the whole system and the sum of that of the NP and clustering adsorbates that are isolated
from each other, divided by $N$, the total number of organic molecules in the simulation cell:
$$\varepsilon = \frac{\varepsilon^{\mathrm{total}} - \left(\varepsilon^{\mathrm{np}} + \sum_{i=1}^{N} \varepsilon_i^{\mathrm{a}}\right)}{N}, \tag{1}$$

where $\varepsilon^{\mathrm{np}}$ is zero for the case without NP.





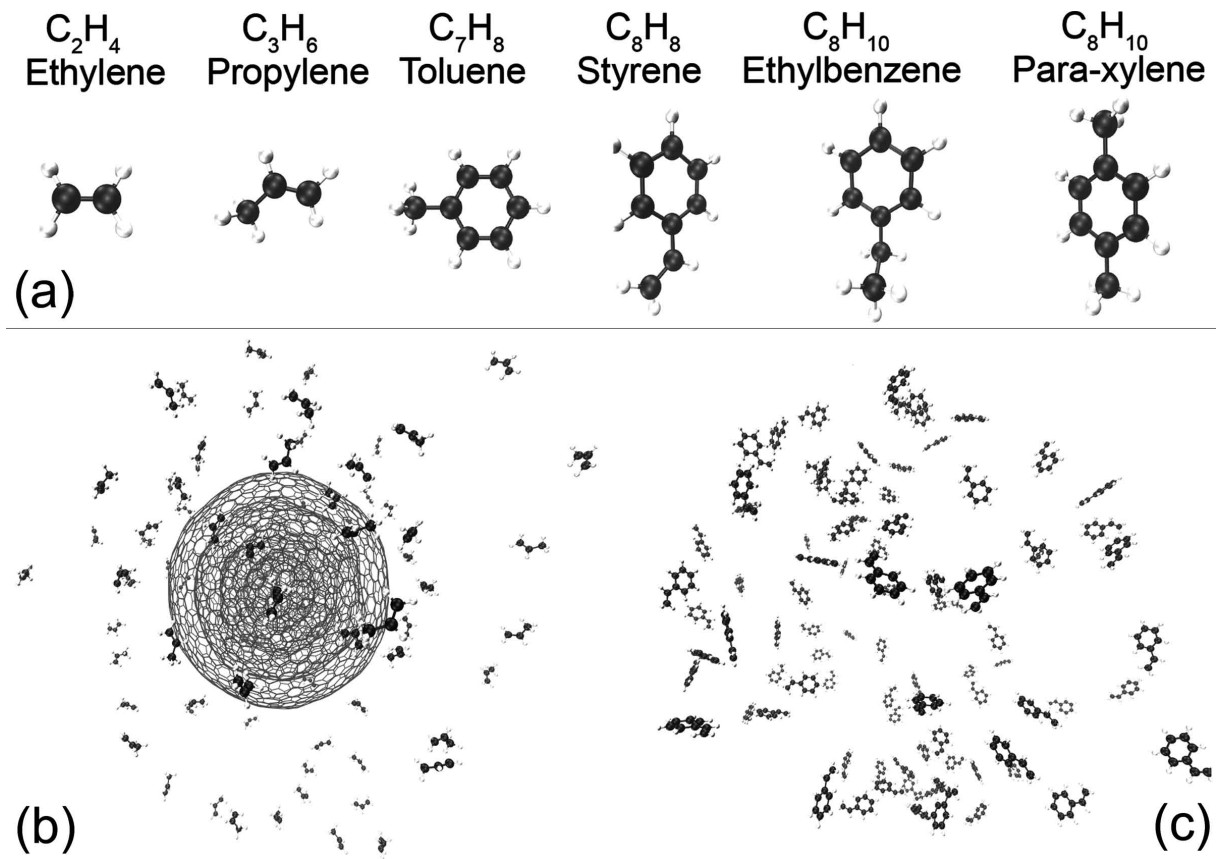

**Figure 1.** (a) Ball-and-stick model of the organic compounds studied in this work. Carbon atoms are depicted in black, hydrogens in gray. (b) and (c) Snapshots of the initial simulation cells for 90 propylene molecules around a 3.64 nm diameter NP, and 90 ethylbenzene molecules without any NP, respectively. This cell is periodic in all three orthogonal directions.

## 3   Results and discussion

Fig. 2 shows the per-molecule binding energy $\varepsilon$ as a function of the number of molecules $N$ with (a), and without (b) a NP in the simulation box. Comparing the two panels, we see that the absolute values of $\varepsilon$ are in general much higher for the case with NP. This means that more energetically-stable clusters can form in presence of the NP which provides a physical substrate of adsorption. Hence, this points to a possible catalytic role of carbon NP in the formation of molecular aggregates. The physisorption of organic compounds could be an important primary stage in the formation of aerosols.

Indeed, positive correlations between the concentrations of aerosol PM and BC have recently been reported by measurements across different continents (Hyvarinen et al., 2011; Marinoni et al., 2010; Ripoll et al., 2014; Rupakheti et al., 2017; Sarkar et al., 2019; Schaap et al., 2004). For instance, a study in a number of European cities has shown that the PM and BC concentrations exhibit similar daily cycles, with a few exceptions caused by secondary formation of particles by means

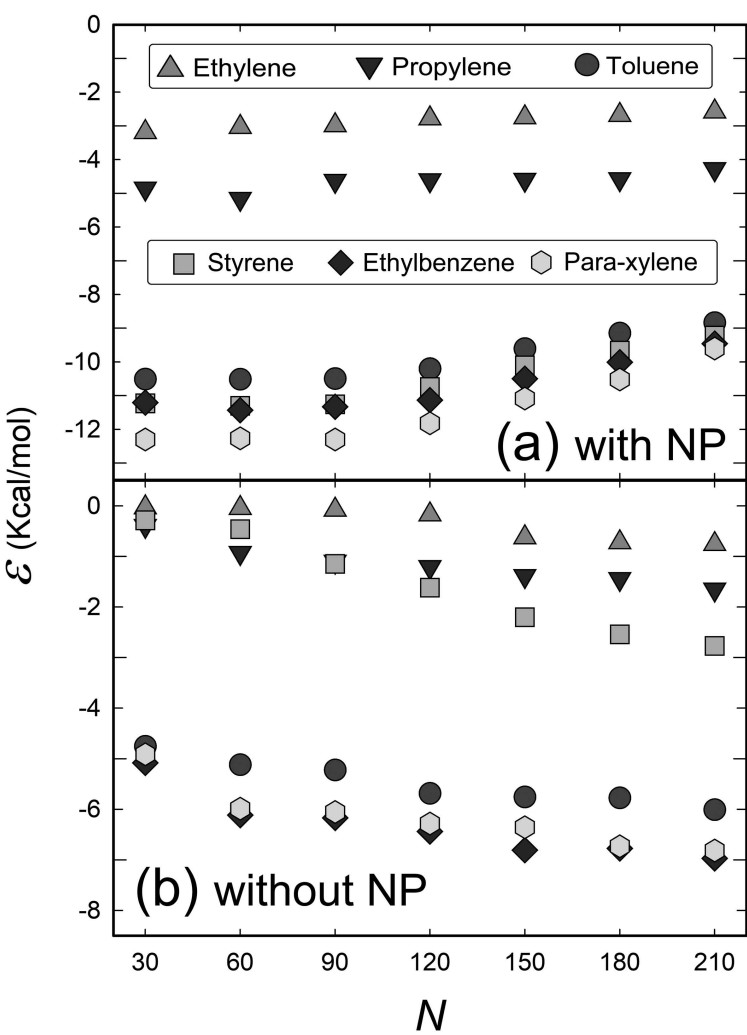

**Figure 2.** Per-molecule binding energy vs. number of molecules for different molecular species (a) physisorbed on the NP surface, or (b) without NP.





of photochemical nucleation processes from gaseous precursors (Reche et al., 2011). Strong correlations between $PM_{2.5}$ mass
and BC concentration have also been observed at urban sites in Korea (Park and Kim, 2004), India (Arif et al., 2018; Marrapu
et al., 2014), New Zealand (Trompetter et al., 2013) and China (Shen et al., 2015; Liu et al., 2019).

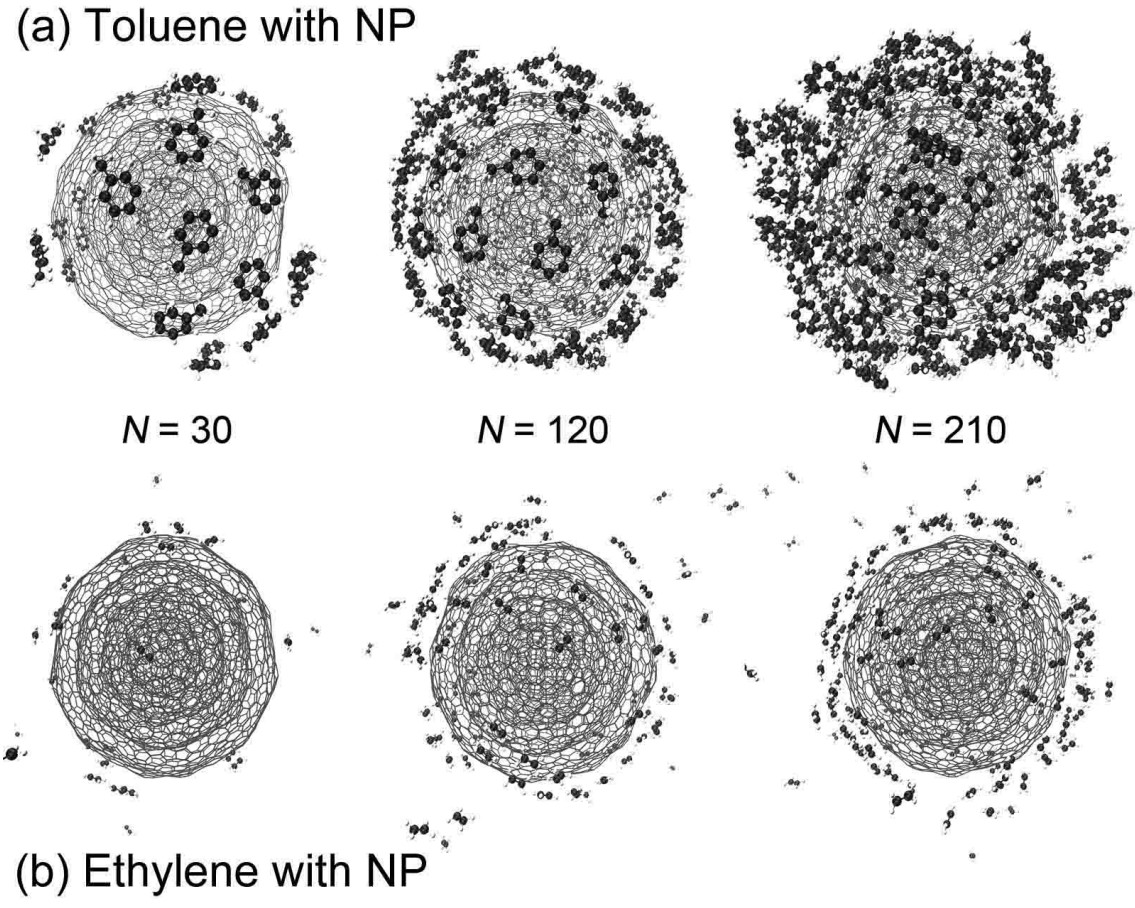

**Figure 3.** Atomistic configurations of different numbers of (a) toluene and (b) ethylene molecules on a NP.

In Fig. 2 (a), the four aromatic compounds (para-xylene, ethylbenzene, styrene and toluene) have clearly higher absolute
values of $\varepsilon$ than the two aliphatic compounds. This is not only due to different numbers of atoms in the molecule, but also
due to the difference in the NP surface coverage. Examining the morphology of the formed clusters, we find that most of the
aromatic molecules aggregate on the surface of NP more readily than the aliphatic ones, as shown in Fig. 3 as examples. For
instance, the right panels ($N = 210$) shows that the toluene molecules start to form three-dimensional (3D) aggregates, while
the ethylene ones form only a thin monolayer with many molecules being isolated in the gas phase. The clusters formed in the
simulations are provided in supplementary data files that contain optimized molecular configurations.
As a general trend, in each simulation, molecules are first observed forming a single thin layer over the NP surface instead
of stacking up in 3D, suggesting a layer-by-layer growth mechanism that is supported by the two slopes of the $\varepsilon$ curves shown
in Fig. 2 (a) (as discussed below). This layer-by-layer growth mechanism could be a general process for aerosol aging in the
atmosphere, since it is consistent with TEM observations of BC embedded within host organic matters in aerosol PM collected
from regions across different continents (Adachi et al., 2010; Katrinak et al., 1993; Li et al., 2003). This clustering process is
clearly illustrated in the animations provided as video supplements.

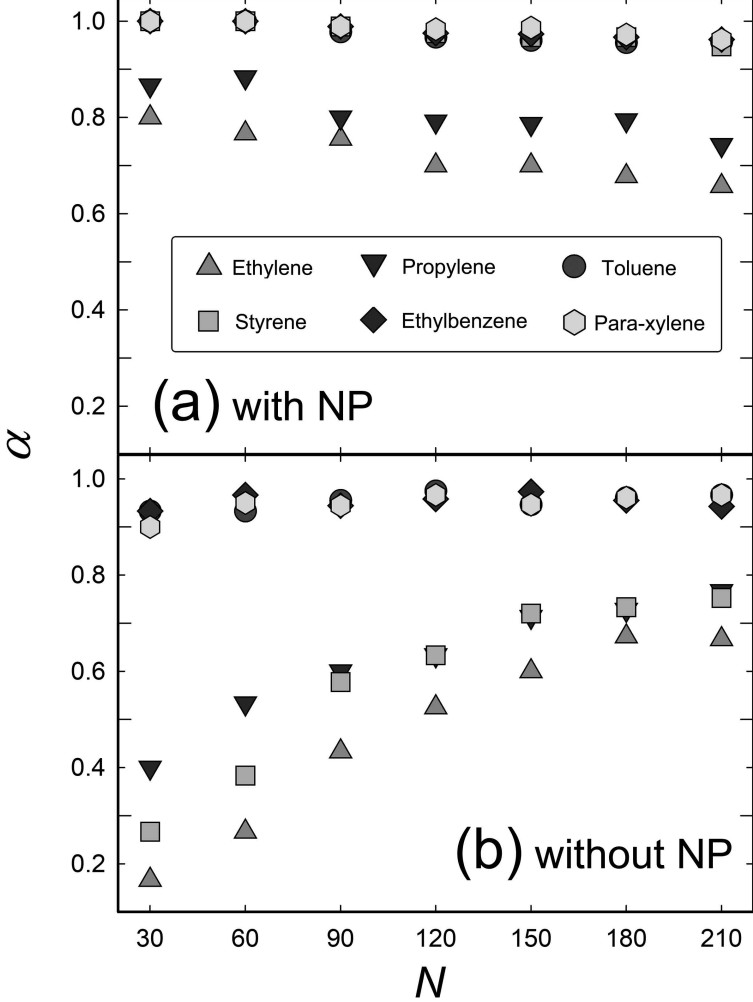

**Figure 4.** Molecular aggregation factor $\alpha$ vs. number of molecules of different molecular species (a) physically adsorbed on the NP surface, or (b) without any NP. $\alpha$ is averaged over thirteen different metastable states.

To quantify how the case shown in Fig. 3 is a general situation in the simulation results, we compute a molecular aggregation
factor for all simulations, which is defined as
$$\alpha = \frac{N - N_{\mathrm{iso}}}{N},$$ (2)





where $N_{\mathrm{iso}}$ is the number of isolated molecules suspended in the gas phase, as defined by a cutoff radius of $1.0\,\mathrm{nm}$ consistent
with the one used in the AIREBO function. $\alpha$ reaches its maximum $1.0$ when all molecules in the simulation box cluster
together and form a single particle. More generally, the higher $\alpha$ is, the larger the molecular aggregates formed in the simulation
box are. In Fig. 4 we plot the values of $\alpha$ as a function of the total number of molecules for different molecular species. A
clear positive correlation is seen between $\alpha$ and the absolute values of $\varepsilon$ shown in Fig. 2. For instance, $\alpha$ values for ethylene,
propylene and styrene are much higher in the presence of the NP than without it, as shown on Fig. 4, and the same trend is
observed in the corresponding values of $\varepsilon$ shown in Fig. 2.
Fig. 4 also shows a difference in the clustering behaviors of aromatic and aliphatic compounds, which leads to different
values of $\varepsilon$ shown in Fig. 2. This difference could come from the particular stacking order of $sp^2$ hybridized carbons, i.e., the
so-called $\pi - \pi$ stacking. Indeed, it is more energetically favorable for an aromatic molecule to be stacked parallel to the NP
surface (Bjork et al., 2010). Hence, the planar structures of aromatic molecules help when forming thermally stable aggregates
on the NP surface. This is consistent with previous results about the selective conduction of organic molecules on the surface
of graphene (Wang, 2019b).
When additional aromatic molecules come to the NP surface after the first thin layer is formed, those molecules are observed
to form $\pi - \pi$ stacks on top of each other and therefore form a core-shell nanostructure. Note that this $\pi - \pi$ stacking occurs
also without NP in the simulation box, as can be seen in Figs. 5 (b) and (c). This selectivity of adsorption suggests that
molecules with planar structures could form more stable and larger aggregates when interacting with BC in atmosphere. This
is in keeping with previously reported correlations between the concentrations of aromatic compounds and aerosol PM, notably
those of polycyclic aromatic hydrocarbons (PAHs) which could have been formed from small aromatic molecules (Haritash
and Kaushik, 2009; Mu et al., 2017; Lyu et al., 2019; Richter and Howard, 2000; Marr et al., 2006; Elzein et al., 2019; Lv et al.,
2016; Polidori et al., 2008).
When there is no NP in the simulation box, the $|\varepsilon|$ for aromatic compounds is also generally higher than for aliphatic ones as
shown in Fig. 2 (b). This difference is consistent with their different clustering behaviors, as evidenced in Fig. 4 (b). Propylene
and ethylbenzene are taken as examples in Figs. 5 (a) and (b) for comparison. Thus, it seems that propylene molecules form
only a few small aggregates, whereas the clusters of ethylbenzene molecules are much larger at the same number density.
An exception, styrene, has much lower values of $|\varepsilon|$ [Fig. 2 (b)] and $\alpha$ [Fig. 4 (b)] in gas phase than other aromatic compounds
of similar molecular structures. Indeed, it is found to be much harder for styrene molecules to form aggregates in the gas phase
than for the three other aromatic species, as shown in Fig. 5 (c). Although the underlying reason for this behavior remains
unclear, we assume that this may come from the $sp^2$-type hybridization of the benzene and the vinyl groups of styrene, so
that both prefer a planar stacking order (Bjork et al., 2010; Kolmogorov and Crespi, 2005). A "comfortable" configuration
might not easily be achieved in gas phase when the styrene molecules are bent due to interaction with their neighbors in the
clusters, evidenced by the fact that the binding energy of styrene is comparable to that of other aromatic compounds with NP
as shown in Fig. 2 (a). By contrast, the $sp^3$ hybridized methyl and ethyl groups in the three other aromatic compounds have
more isotropic stacking orders.





*N* = 30    *N* = 120    *N* = 210

(a) Propylene without NP

(b) Ethylbenzene without NP

(c) Styrene without NP

**Figure 5.** Atomistic configurations of different numbers of (a) propylene, (b) ethylbenzene and (b) styrene molecules without any NP.





## 4 Conclusions

The physisorption of six organic compounds on the surface of a carbon NP is simulated in order to mimic the primary formation stage of aerosols with VOC precursors. The results of our binding energy calculations show that more stable clusters can form thanks to the presence of the NP, and thus point to a catalytic role of BC in the formation of aerosol PM. This could be useful for understanding the correlation between experimentally-measured concentrations of aerosol PM and BC. It is also found that the absolute binding energy of the aromatic compounds is different from that of aliphatic ones, due to the large difference in their clustering behaviors. This could be related to previously-reported correlation between the concentrations of aromatic compounds and aerosol PM, in particular PAHs. Furthermore, analysis on the morphology of the resulting particles points to a layer-by-layer formation process of aerosol PM in atmospheric aging, in agreement with experimental observations of BC embedded within host organic matters in aerosol PM.

*Code availability.* Sets of simulation input scripts are available via https://dx.doi.org/10.5281/zenodo.3628331 so that interested readers can repeat the simulations.

*Data availability.* Data files that contain optimized atomistic configurations of organic molecules adsorbed on nanoparticles are provided via https://dx.doi.org/10.5281/zenodo.3628331. These .xyz files contain the atomic coordinates of the adsorbed organic molecules. The first line in each file contains the total number of atoms, the second line comprises three integers corresponding to the number of molecules, number of atoms in each molecule and number of atoms in the nanoparticle, and each subsequent line contains the atomic species and the three Cartesian coordinates (in Å) for an atom. Please see the PDF in the zipped file for instructions.

*Video supplement.* Video supplements are available via https://dx.doi.org/10.5281/zenodo.3628331 for demonstrating the formation process of molecular clusters.

*Author contributions.* Z. W. conceived and designed the simulations. X. Z. performed the simulations and collected the data. Z. W., X. Z., Y. Z., S.P. and M. D. contributed data analysis. Z. W., Y. Z., S. P., M. D., J. C. and G. M. wrote the paper.

*Competing interests.* The authors declare no competing interest.



170  *Acknowledgements.* Peter Blaha and Karlheinz Schwarz are acknowledged for helpful discussions. Partial financial supports from the Na-

171  tional Natural Science Foundation of China (11964002), the Guangxi Science Foundation (2018GXNSFAA138179) and the Scientific Re-

172  search Foundation of Guangxi University (XTZ160532) are acknowledged.





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
