# Peer review of "Role of black carbon in the formation of primary organic aerosols: Insights from molecular dynamics simulations"

_Atmospheric Chemistry and Physics, 2020_

## Referee Comment (RC1) · Xiaoxiang Wang (Referee) · 3 Mar 2020

This succinct paper used molecular dynamics simulations to investigate the physisorption of gaseous organic molecules on the surface of a carbon nanoparticle. The pollutant molecules were found to cluster in different ways in the presence of the NP than in the gas phase. Authors also suggested a layer-by-layer formation process of aerosol PM, consistent with the onion-like nanostructures of aerosol particles observed before. The topic studied is highly related to the scope of ACP, and results found should be interesting for atmospheric researchers. I suggest the paper is accepted for publication in ACP after some minor issues are addressed.

[Figure]

General comments:

1. Most importantly, it would be nice to see more discussions about the implications of this study for our atmospheric environment. Readers of ACP will like these contents.

2. Figures in the paper are monochrome. I am not sure if it is the problem of submission system. Colored pictures are preferred.

3. Authors used a bucky-onion of 3.64 nm in diameter containing four concentric fullerene layers to represent the carbon nanoparticle in the simulation. The reason of choosing this size was also given: "The system size is kept small due to computational cost considerations, even though BC particles in urban atmospheres can grow from tens to over a hundred nanometers after mixing with other compounds through atmospheric aging processes". As the size of nanoparticles play a key role in the behavior of particles in the sub-10 nm region, I suggest authors to do a series of simulations: one species of organic molecule on the surface of particles with different diameters. Authors can just do simulation and check if the findings based on 3.64 nm particles are still working. If authors do not like to do it, please make it clear why in the paper.

4. In line 80, authors claimed that "A key coefficient influencing the clustering of molecules, the per-molecule binding energy $\varepsilon$ is calculated". I suggest authors give more descriptions about this parameter, e.g. explaining why you choose it.

---

## Referee Comment (RC2) · Xiaoxiang Wang (Referee) · 16 Mar 2020

Dear Authors,

Yes, you can address these issues after the epidemic.

Please contact the editorial office if you need an extension of "review and discussion".

Best regards, Xiaoxiang
* * *

---

## Short Comment (SC1) · 16 Mar 2020

Dear Referee Xiaoxiang Wang,

Thank you for your insightful comments that will greatly improve the manuscript. Unfortunately we are not able to provide more detailed response for the moment, due to a temporary close-down of our labs in the consequence of the coronavirus outbreak. We will revise our manuscript as soon as our computational facilities become accessible. This is in particular important for answering your comment No.3 to perform simulations with a nanoparticle of different size, as well as for the comment No.2. All of your suggestions will be included in the next manuscript revision along with a detailed

point-to-point response.

Yours faithfully, on behalf of the co-authors.

Zhao Wang

---

## Referee Comment (RC3) · Anonymous Referee #2 · 7 Apr 2020

This manuscript examines the estimated binding energies between a soot nanoparticle and volatile organic compound vapors compared with the self-binding energies among the organic vapors through a series of molecular dynamics simulations. While the simulations and energy estimation methods appear to be technically sound, the proposed relevance to atmospheric processes does not seem to be supported. The manuscript is thus recommended for rejection from ACP at this time.

General comments:

The main conclusion advocated by the authors is that VOC adsorption onto black carbon nanoparticles (NPs) plays an important role in formation of PM from VOCs be-

cause their binding energies to NPs are greater than their self-binding energies.

One confusing aspect of the manuscript is that the authors talk about aerosol nucleation, but the carbon NPs are already in aerosol form - so condensation of new material is not necessarily nucleating new particles. It could be argued that a new, condensed organic phase is being created/expanded via gas/particle partitioning (and thus contribute to initial condensation/"atmospheric aging") but the motivating discussion is not phrased this way.

The authors suggest that past correlations observed between BC and PM are due to the gas/particle partitioning mechanism they propose, but the correlation is more likely due to meteorology (varying boundary layer height) and transport (polluted air masses tend to contain both). And a large fraction of PM is inorganic, which is unexplained by the authors' hypothesis.

The binding energies the authors have compared are related to the adsorption energies and (free) activation energies for homomolecular nucleation. The atmospheric implications of the molecular dynamics calculations are better placed in context by incorporating their estimates into appropriate adsorptive gas/particle partitioning or nucleation models with atmospherically-relevant concentrations. However, it is generally regarded that homomolecular nucleation is not a likely mechanism for new particle formation in the atmosphere (based on past studies of binding energies and concentrations), so comparison of NP-binding energies to self-binding energies - particularly of VOCs - is not likely to yield a meaningful reference point on the significance of the former. The difference in binding energies though do support the layered mechanism of condensation observed in this work rather than organic island formation shown in many past molecular dynamics simulations of nanoparticles. (However, the universality of layered particles with soot at the center - i.e., core-shell morphology - has also been challenged by experimental data in the last decade.)

To determine the importance of the calculated NP-binding energies, the authors may

refer to different adsorptive partitioning models - e.g., Pankow 1987 (reference below). On a mass basis, this mechanism (adsorptive partitioning) is generally considered to be less important overall than absorptive partitioning (also see articles by Pankow and co-workers on the topic). However, for certain toxic compounds, estimating the gas/particle partitioning behavior from computational simulations can prove useful and the authors may wish to direct their efforts in this area. (For analysis of experimental data for adsorptive/absorptive partitioning of aromatic and aliphatic organics, see also Pankow and co-workers' publications from the 90's.)

Pankow, J.F., 1987. Review and comparative analysis of the theories on partitioning between the gas and aerosol particulate phases in the atmosphere. Atmospheric Environment (1967) 21, 2275–2283. https://doi.org/10.1016/0004-6981(87)90363-5

Regarding the NP-binding energies, the simulation of a carbon NP is not entirely unreasonable since "fresh soot" is typically comprised of small, agglomerated spherules, albeit at varying sizes. However, the issue of sensitivity to surface curvature raised also by the other reviewer remains an impediment for interpreting the greater relevance of the current work. Extending the concept of the Tolman length to solids, there should be a size limit beyond which the curvature affects the results minimally, in the range where many carbon spherules and soot particles are found. Would not a comparison against binding to a flat graphene surface help alleviate this question?

Minor comments:

How large is the estimated binding energy compared to 1) variability among the local minima and 2) atomization energy of the BC itself?

Since an annealing process is used prior to the binding energy calculation, what temperature do these energies correspond to?

The word "catalytic" seems to be used colloquially but can be confusing in a chemistry-oriented paper.

The authors may wish to consider or discuss the conventional wisdom that soot is typically emitted with defects, surface functionalization, and coating by organic lubricants, etc. (I believe one of the authors of this manuscript has experience simulating adsorption on functionalized graphitic surface for this reason).

LAMMPS, as an acronym for Large-scale Atomic/Molecular Massively Parallel Simulator, is usually capitalized.

---

## Author Comment (AC1) · 13 May 2020

Dear Dr. Manabu Shiraiwa,

Thank you for sending us the comments of the reviewers, and for your understanding regarding the delay in our reply due to the pandemic. Those comments have greatly helped to improve the quality of the manuscript.

Point-by-point answers to the referees are included below, in the following se-

[Figure]

quence: (1) comments from Referees, (2) author's response, (3) author's changes in manuscript. For convenience, our response is typeset in blue font.

We are looking forward to hearing from you in due course, and thank you for your consideration.

On behalf of all the co-authors,

Sincerely,

Prof. Zhao Wang

Xiaoxiang Wang (Referee #1)
Comments:
This succinct paper used molecular dynamics simulations to investigate the physisorption of gaseous organic molecules on the surface of a carbon nanoparticle. The pollutant molecules were found to cluster in different ways in the presence of the NP than in the gas phase. Authors also suggested a layer-by-layer formation process of aerosol PM, consistent with the onion-like nanostructures of aerosol particles observed before. The topic studied is highly related to the scope of ACP, and results found should be interesting for atmospheric researchers. I suggest the paper is accepted for publication in ACP after some minor issues are addressed.

Response: We thank the Referee for their remarks on our manuscript clearly falling within the scope of ACP and for providing these insightful comments. We have made appropriate changes to the manuscript, as described in the point-by-point response below.

General comments: 1. Most importantly, it would be nice to see more discussions about the implications of this study for our atmospheric environment. Readers of ACP will like these contents.

Response: We have expanded the discussion in response to this comment. The paragraph starting at line 91 has been expanded as follows:

"Our molecular simulations suggest that the BC could provide effective adsorption sites for organic molecules, which help to facilitate the initial growth of aerosol particles. This could provide a new microscopic mechanism for the positive correlations between the concentrations of aerosol PM and BC that have recently been reported by measurements across different continents (Hyvarinen et al., 2011; Marinoni et al., 2010; Ripoll et al., 2014; Rupakheti et al., 2017; Sarkar et al., 2019; Schaap et al., 2004; Chen et al., 2016). For instance, a study in a number of European cities has shown that the

PM and BC concentrations exhibit similar daily cycles, with a few exceptions caused by secondary formation of particles by means of photochemical nucleation processes from gaseous precursors (Reche et al., 2011). In another long-term study at eastern Himalaya, BC concentrations have found to be highly correlated with PM2.5 during the post-monsoon season, whereas the correlation was weaker during the pre-monsoon season which could be attributed to the long-distant transport of dust aerosols and the formation of secondary particle (Sarkar et al., 2019). Strong correlations between PM2.5 mass and BC concentration have also been observed at urban sites in Korea (Park and Kim, 2004), India (Arif et al., 2018; Marrapu et al., 2014), New Zealand (Trompetter et al., 2013) and China (Shen et al., 2015; Liu et al., 2019). In a comparative study of a roadside station and another site relatively farther away from the highway, the BC factor has been determined to be the major cause of the PM2.5 imbalance between the sampling locations (Sofowote et al., 2018). It was widely recognized that the behavior of PM and BC depends mainly on the type of emission sources, the meteorological conditions and the geographical factors. The good correlation between PM and BC suggested that they have similar emission sources and transformation trends (Shen et al., 2015; Park and Kim, 2004; Sarkar et al., 2019; Gatari et al., 2019).

Furthermore, the following discussion has been added after line 126:

Moreover, according to the calculations based on a multi-component kinetic model in which the formation and growth of clusters were regarded as continuous collisions and selective aggregations of molecules (Xia et al., 2016; Jiang et al., 2018), it has been found that the cyclic VOCs contribute the most to the aerosol particulate formation as compared to the linear and branched VOCs (Jiang et al., 2019).

The following sentence was added in line 16:

"and could provide some new insight into the interpretation of the experimental measurements on aerosol PMs and VOCs."

2. Figures in the paper are monochrome. I am not sure if it is the problem of submission system. Colored pictures are preferred.

Response: We have replaced Figures 1, 3, 5 by color versions in response to this comment. We had previously chosen monochrome figures because of their small size (allowing for higher resolutions).

3. Authors used a bucky-onion of 3.64 nm in diameter containing four concentric fullerene layers to represent the carbon nanoparticle in the simulation. The reason of choosing this size was also given: "The system size is kept small due to computational cost considerations, even though BC particles in urban atmospheres can grow from tens to over a hundred nanometers after mixing with other compounds through atmospheric aging processes". As the size of nanoparticles play a key role in the behavior of particles in the sub-10 nm region, I suggest authors to do a series of simulations: one species of organic molecule on the surface of particles with different diameters. Authors can just do simulation and check if the findings based on 3.64 nm particles are still working. If authors do not like to do it, please make it clear why in the paper.

Response: We thank the Referee for pointing out this important issue. In the revision, we have obtained results of simulations with a larger carbon nanoparticle (diameter = 4.58 nm) for comparison, as shown in the panel (c) of the Fig.1 at the end of this file below.

It can be seen from the comparison between the panels (b) and (c) that the binding energy of the molecules on the larger NP exhibits similar orders for different species. Although both the panels (b) and (c) show that the energy curve changes the slope with increasing number of the adsorbed molecules, the breaking-point is different for the small and large NPs. For instance, the energy-N curves of toluene, styrene, ethylbenzene and para-xylene change approximately the slope at N=90 on the small NP [panel (b)], while those do so at N=150. This change originates from the fact that

the molecules interact more strongly with the NP than with other molecules, and the change of the breaking point with NPs of different sizes is correlated with the surface coverage. In our simulations, when a small number of molecules are adsorbed, they are observed to distribute homogeneously on the nanoparticle surface, forming a thin monolayer. The binding energy is roughly a linear function of the number of adsorbed molecules before the surface is saturated. After the saturation, molecules start to stack up to form 3D aggregates, which lead to another linear increase of the adsorption energy with lower magnitude of the per-molecule energy. This Figure and the above discussion are included in the manuscript in Pages 5-7.

4. In line 80, authors claimed that "A key coefficient influencing the clustering of molecules, the per-molecule binding energy " is calculated". I suggest authors give more descriptions about this parameter, e.g. explaining why you choose it.

Response: In response to this comment, we have added the following sentence to the end of Page 3. "The magnitude of the binding energy is a direct measure of the interaction between molecules, and thus indicates how easy or difficult it is for these molecules to form aggregates."

Anonymous Referee #2
This manuscript examines the estimated binding energies between a soot nanoparticle
and volatile organic compound vapors compared with the self-binding energies among
the organic vapors through a series of molecular dynamics simulations. While the sim-
ulations and energy estimation methods appear to be technically sound, the proposed
relevance to atmospheric processes does not seem to be supported. The manuscript
is thus recommended for rejection from ACP at this time.

Response: We thank the Referee for the constructive comments, but respectfully dis-
agree on his/her general point about relevance. Our results are meaningful and can
be used by atmospheric researchers, an opinion also supported by the comments of
Referee #1. We explain why in the point-to-point response to the comments of Referee
**2 below.**

General comments: The main conclusion advocated by the authors is that VOC ad-
sorption onto black carbon nanoparticles (NPs) plays an important role in formation
of PM from VOCs because their binding energies to NPs are greater than their self-
binding energies. One confusing aspect of the manuscript is that the authors talk about
aerosol nucleation, but the carbon NPs are already in aerosol form - so condensation
of new material is not necessarily nucleating new particles. It could be argued that a
new, condensed organic phase is being created/expanded via gas/particle partitioning
(and thus contribute to initial condensation / "atmospheric aging") but the motivating
discussion is not phrased this way.

Response: This manuscript is mainly oriented to the experimental audience, so we
have chosen terms commonly used by experimentalists. Since carbon NPs are usually
observed to be enveloped by organic matter in the atmosphere, the organic aerosols
in this manuscript refer to these mixed particles. As a matter of fact, we had already
opted for the term "clustering" instead of "nucleation" in most parts of our original
submission to avoid confusing readers, as chemical reactions are not studied in this
work. Despite the variety of the language used in the field, we hesitated to use the

term "aging", which refers to long-term processes resulting in microstructures more complex than the ones simulated in this work.

The authors suggest that past correlations observed between BC and PM are due to the gas/particle partitioning mechanism they propose, but the correlation is more likely due to meteorology (varying boundary layer height) and transport (polluted air masses tend to contain both). And a large fraction of PM is inorganic, which is unexplained by the authors' hypothesis.

Response: The present manuscript provides a comparison of the binding energies between the clustering of different gas compounds and carbon NP, and proposes a new microscopic mechanism that influences the formation of the fine aerosols. This is obviously a single aspect of the problem, and many other factors could be involved in the long-term atmospheric aging at the meso- and macroscopic scales. Our results have no conflict with the meteorology and transport mechanisms. We understand that realistic conditions could be much more complex and that the ultimate proof of this effect will come from experiments, not from simulations. However, detailed analysis of individual factors is not only useful, but a fundamental step in disentangling such complex processes. For instance, as mentioned by the Referee, a large fraction of PM is inorganic, which are out of the scope of the present manuscript that focuses on organic aerosols. However, it is worth noting that the gas/particle partitioning of organic molecules, especially aromatic ones, is a very important question. Similarly, the role of soot particles as an efficient substrate for adsorption of these organic molecules has long been recognized (e.g., Dachs and Eisenreich, Environ. Sci. Technol. 2000, 34, 3690; Ma et al, 2019, Sci. Tot. Environ. 693, 133623; Gaga and Ari, Atmos. Pollut. Res. 2019, 10, 1). In connection with these topics, molecular dynamics simulations offer a unique tool to investigate the behavior of organic species by considering different molecules separately, i.e., by isolating the effect of one or a few the molecular properties (aromatic vs aliphatic, length of the aliphatic chain, internal geometry...).

The binding energies the authors have compared are related to the adsorption energies and (free) activation energies for homomolecular nucleation. The atmospheric implications of the molecular dynamics calculations are better placed in context by incorporating their estimates into appropriate adsorptive gas/particle partitioning or nucleation models with atmospherically-relevant concentrations. However, it is generally regarded that homomolecular nucleation is not a likely mechanism for new particle formation in the atmosphere (based on past studies of binding energies and concentrations), so comparison of NP-binding energies to self-binding energies - particularly of VOCs - is not likely to yield a meaningful reference point on the significance of the former. The difference in binding energies though do support the layered mechanism of condensation observed in this work rather than organic island formation shown in many past molecular dynamics simulations of nanoparticles. (However, the universality of layered particles with soot at the center - i.e., core-shell morphology - has also been challenged by experimental data in the last decade.)

Response: The simulations of homomolecular clustering are important to perform comparisons between the binding energies of different gas compounds, which can help tell which species can adsorb more than others, and hence might be more "polluting", despite being qualitatively, rather than quantitatively, descriptive of realistic situations.

To determine the importance of the calculated NP-binding energies, the authors may refer to different adsorptive partitioning models - e.g., Pankow (reference below). On a mass basis, this mechanism (adsorptive partitioning) is generally considered to be less important overall than absorptive partitioning (also see articles by Pankow and co-workers on the topic). However, for certain toxic compounds, estimating the gas/particle partitioning behavior from computational simulations can prove useful and the authors may wish to direct their efforts in this area. (For analysis of experimental

data for adsorptive/absorptive partitioning of aromatic and aliphatic organics, see also Pankow and co-workers' publications from the 90's.) Pankow, J.F., 1987. Review and comparative analysis of the theories on partitioning between the gas and aerosol particulate phases in the atmosphere. Atmospheric Environment (1967) 21, 2275–2283. https://doi.org/10.1016/0004-6981(87)90363-5

Response: We thank the Referee for pointing us in this interesting direction. Based on linear Langmuir isotherms, the Junge-Pankow adsorption model relates the fraction of chemicals in the particle phase with the sub-cooled liquid vapor pressure of the pure compound and the particle surface area. This model has been proven successful in predicting the gas/particle partitioning behavior of SVOCs. The atomistic model used in this work provides dynamic details about the molecular interactions of specific compounds at a fundamental level, and the results could be more general. We think that both methods are useful for atmospheric research and that their results could be complementary to each other. Incorporating the results of atomistic simulations into classical adsorptive gas/particle partitioning models could be very interesting for developing a new multi-scale approach. We have added the following discussions about the Junge-Pankow models in the conclusion of the revised manuscript, and definitively agree with the Reviewer that we should look for collaborations in the future with experts in this area.

"The present MD simulations provide a molecular-resolution view of the physisorption processes of gaseous organic molecules. Such gas/particle partitioning behavior can also be modeled using the Junge-Pankow adsorption model (Pankow, 1987) which is based on linear Langmuir isotherms. In our future work, it will be interesting to incorporate the results of atomistic simulations into classical adsorptive gas/particle partitioning models for developing a new multi-scale approach."

Regarding the NP-binding energies, the simulation of a carbon NP is not entirely unreasonable since "fresh soot" is typically comprised of small, agglomerated spherules,

albeit at varying sizes. However, the issue of sensitivity to surface curvature raised also by the other reviewer remains an impediment for interpreting the greater relevance of the current work. Extending the concept of the Tolman length to solids, there should be a size limit beyond which the curvature affects the results minimally, in the range where many carbon spherules and soot particles are found. Would not a comparison against binding to a flat graphene surface help alleviate this question?

Response: We agree with the Referee that the issue of sensitivity to surface curvature must be clarified. In the re-submission, we have performed simulations with a different particle size. New results and discussion were added to the revised manuscript. Please see the response to the 3rd comment of the Referee #1 above.

Minor comments: How large is the estimated binding energy compared to 1) variability among the local minima and 2) atomization energy of the BC itself?

Response: 1) The averaged relative root mean square deviation with respect to the value show in Figure 2 is about 3.213% for the case with NP and is about 30.59% for the pure gas phase. 2) The atomization energy of the carbon NP itself is about -22294 eV for 3140 atoms in total (about -7.1 eV per atom), as the formation energy of a sp2 carbon bond is orders of magnitude larger than that of the vdW interaction in vacuum.

Since an annealing process is used prior to the binding energy calculation, what temperature do these energies correspond to?

Response: The binding energy was calculated from the annealed ground-state, which is from a "quickly-frozen" picture of the system at 300 K.

The word "catalytic" seems to be used colloquially but can be confusing in a chemistry oriented paper.

Response: We thank the Referee for this comment and have removed all mentions of the words in favor of a more precise description of the process we deal with.

The authors may wish to consider or discuss the conventional wisdom that soot is typically emitted with defects, surface functionalization, and coating by organic lubricants, etc. (I believe one of the authors of this manuscript has experience simulating adsorption on functionalized graphitic surface for this reason).

Response: Following this comment, we will add discussion about defects in soot into the manuscript. For instance: There must be many defects on the surface of soot, as shown by electronic microscopy [ Parent 2016 https://doi.org/10.1016/j.carbon.2016.01.040 ]. These defects could be oxygenated functional groups (mainly C=O) and/or carbon atom vacancies and/or structural 5-7 defects (the so-called "Stone-Wales" defects). However, both the type and density of these defects depend on many factors such as the fuel used, the combustion conditions and the residence time of the soot nanoparticles in the atmosphere. It would be thus very challenging to make a modeling of soot nanoparticles in real conditions. Thus, modeling adsorption on a perfect surface as in the present paper, can be considered as the first, and unavoidable step for further studies.

LAMMPS, as an acronym for Large-scale Atomic/Molecular Massively Parallel Simulator, is usually capitalized.

Response: We thank the Referee for this comment and have replaced "Lammps" by "LAMMPS".
* * *
[Figure]

Fig. 1.